# External Confined Concrete Cylinders Behavior under Axial Compression Using CFRP Wrapping

**DOI:** 10.3390/ma15228232

**Published:** 2022-11-19

**Authors:** Abdelhamid Karouche, Kamel Hebbache, Cherif Belebchouche, Noureddine Lahbari, Oussama Kessal, Slawomir Czarnecki

**Affiliations:** 1LGC-ROI, Department of Civil Engineering, Faculty of Technology, University of Batna 2, Batna 05000, Algeria; 2Department of Civil Engineering, Ferhat Abbas University of Setif 1, Setif 19000, Algeria; 3Department of Civil Engineering, Faculty of Sciences of Technology, Mentouri Brothers University of Constantine 1, Constantine 25000, Algeria; 4Department of Civil Engineering, Faculty of Science and Technology, University of Bordj Bou Arriridj, El Anceur 34000, Algeria; 5Department of Materials Engineering and Construction Processes, Wroclaw University of Science and Technology, Wybrzeze Wyspianskiego 27, 50-370 Wroclaw, Poland

**Keywords:** compressive strength, sand/resin effect, water–cement ratio, confinement effect, volumetric ratio, CFRP

## Abstract

Carbon-fiber-reinforced polymer (CFRP) is a composite material used to mend and strengthen concrete structural elements in civil engineering. The prime aim of this experimental study is to investigate the comportment of confined concrete cylinders (CCC) under uniaxial loads by altering the concrete strength, the CFRP angle orientation, and the volumetric ratio, following the externally bonded reinforcement technique (EBR). We present the results of the confinement effect and failure mechanisms issue of more than 150 specimens of CFRP confined concrete cylinders that have been undertaken and in which several parameters were altered. Totally and partially confined concrete cylinders were tested for failure under axial compressive loads and indirect tensile tests. Four different ratios of water/cement (0.33, 0.36, 0.401, and 0.522) were investigated. In addition, three sand–resin ratios were prepared to improve the mechanical properties and the adhesion of the CFRP and the concrete. The obtained results revealed a clear improvement in the compressive strength of the specimens made with low strength concrete (from 38% to 66%) compared to those made of high strength concrete (from 11% to 31%), where the improvements are relatively low. Furthermore, the transversally confined concrete cylinders presented significant gains in strength over those confined longitudinally. Lastly, adding sand to the resin increases the compressive strength of confined concrete cylinders (1.19% to 54.62%) and reduces the cost of the resin used for fixing CFRP materials.

## 1. Introduction

Concrete is the most used material in the world of construction. This material can be affected by internal and external actions (freeze–thaw cycle, alkali–silica reaction, carbonation, etc.). These actions can decrease the carrying capacity of the concrete structural elements. Over the last four decades, the strengthening of reinforced concrete has significantly developed. The benefits of reinforcement are essentially focused on conserving the environment. The reinforcement of concrete structures postpones their demolition. Waste debris is noisy and dusty and may produce hazardous materials for adjacent buildings that require removal. Furthermore, to preserve the archaeological heritage for future generations, there is a need to increase the performance of the load-bearing elements of old structures.

Carbon-fiber-reinforced polymer (CFRP) is a composite material to strengthen concrete structures. Several reinforcement techniques based on various materials have been reported in the literature, using carbon- or glass-fiber-reinforced polymer (CFRP or GFRP) and steel plates. Structural elements, such as the beams [1,2,3], columns [4,5,6,7,8], beam/column joints [9,10,11], and the slabs [12,13,14] were among the studied subjects. These reinforcements demonstrated the carrying capacity enhancement of the strengthened elements.

As the concrete weakens in sustaining tensile loads, it starts to crack and eventually crumbles and disintegrates at even weaker stresses. Structural element confinement is the best-suited technique against future hazardous actions to overcome this behavior. Columns’ reinforcement using composite materials is carried out by wrapping them with FRP composite. The lateral expansion of concrete under compressive stress leads the tensile on the reinforcement in the fibers’ way. As a result, the confinement pressure increases proportionally with the lateral expansion until the composite material fails.

The main aim of the present experimental work is to study the behavior of the confined concrete cylinders (CCC) by using the CFRP composite under uniaxial load using the following parameters: the concrete strength (from 25 MPa up to 47 MPa), the CFRP angle orientation (0° and 90°), and the volumetric ratio (from 0.46% to 2.5%). In addition, the aim is also to check the obtained results by Abdulla et al. [15] when adding sand to resin to use it as a sealing material of a CFRP strip in concrete columns. Moreover, a statistical modelling represented by the variance analysis was adopted in order to calibrate the obtained results with the predicted values.

## 2. Literature Review

Table 1 presents recent studies on reinforced cylinders and columns in which CFRP composites were utilized. The Table 1 also indicates the specific studied parameters and the obtained results.

According to this literature review, the most researched parameters were the volumetric ratio, the CFRP layer number, the slenderness ratio, the spacing between the band fibers, the concrete strength, and the CFRP orientation angle. As such, several issues need to be clarified: (1) the behavior of partially confined cylinders with an orientation angle of 90°; (2) the confinement ratio effect on the tensile strength; and (3) the failure mode of partially confined cylinders having an orientation angle of 90°.

It is worth mentioning the work of Abdulla et al. [15], where they investigated the enhancement of mechanical and thermal properties of resin by adding a granular class that ranges from 75 μm to 300 μm to fix CFRP strips on beams. They concluded that when the resin types with sand to resin (S/R) ratio reach the unit, this improves the concrete properties. However, the use of this kind of resin to fix CFRP wrap has not been studied before and remains poorly understood to evaluate the behavior of reinforced concrete cylinders; hence its study is essential to confirm the work initiated by Abdulla et al. [15].

## 3. Experimental Details

### 3.1. Materials

A total of 150 concrete specimens were examined in this study. Calcareous crushed aggregates from the quarry of Ain Roua region of Sétif, Algeria, were used. They are of three granular classes: sand 0/5 mm, gravel 5/15 mm, and gravel 15/25 mm (Figure 1). Ordinary Portland cement (CEM II 42.5) was used for casting the specimens. This cement comes from the Ain El kebira region of Sétif, Algeria and its physico-chemical characteristics are presented in Table 2. The design procedure consisted at first of mixing crushed coarse aggregates, crushed sand, and cement constituents. Then, the water was added to the mixture design. After that, the concrete was poured into three layers and vibrated with a vibrator. Finally, molds with dimensions of 16 × 32 cm^2^ (diameter × length), in order to obtain a desirable model of failure [31], were filled with the excess concrete removed, and the upper surface was leveled and scraped. This procedure is similar to that described by the Dreux–Gorisse method [32]. More details of the mixtures are presented in Table 3. Keeping the same granular skeleton = decreased the water to cement ratio (W/C) to improve the concrete strength and avoid workability.

Furthermore, a superplasticizer (SP) was added depending on the cement weight (see Table 3) to improve the workability and to be able to produce an acceptable slump (plastic or highly plastic concrete according to EN 12350-2). Prior to the preparation of the mixtures’ design, physical and mechanical characterization tests were performed on all concrete components, including sieve analysis (SA) [33], the fineness modulus (FM = 3.2) [34], Los Angles (LA = 20%) [35] and the sand equivalent test (SE = 74%) [36]. In addition, all physical characterizations, specimen preparations, compressive cylinder strength [37], and indirect tensile strength [38] tests were carried out in the construction material laboratory (UFAS1). The mix design and the main properties of concrete types are summarized in Table 3.

The carbon-fiber-reinforced polymer (CFRP) used in this experimental study was a 230C/45 type Sika Wrap. The used CFRP type was unidirectional, manufactured by SIKA Company (https://dza.sika.com/, accessed on 8 June 2021). The epoxy adhesive used in this study was (Sikadur^®^-330), composed of two components: resin and hardener, which were mixed in a ratio of 1/4. The main features of epoxy and CFRP wraps are summarized in Table 4.

### 3.2. Fabrication of Test Specimens

The carbon-fiber-reinforced polymers (CFRP) sheets were fixed on the specimens using an epoxy adhesive (Sikadur^®^-330). To ensure a maximal bond between CFRP and concrete, the contact surfaces of the cylindrical concrete specimens were raised and brushed using a stainless-steel brush. Then, a compressed air (air gun) was used to remove any impurities and ensure perfect adhesion between the epoxy–concrete and the epoxy–CFRP. Next, the CFRP sheets were cut in bands of different lengths and widths to cover the required surface on the concrete specimens. For strengthening the cylindrical concrete specimens, an epoxy layer at about 1 mm of thickness was applied on the surface of the concrete and the carbon sheets. Then, a slight pressure was applied by a roller to wipe out voids. Moreover, another layer of epoxy was applied to the CFRP sheets. Finally, the concrete specimens were placed in the open air at an ambient temperature of 25 °C inside the workshop of the laboratory until the testing day.

Horizontal and vertical confinement configurations were adopted. The first one consisted of confining the specimens horizontally using three confinement ratios (25%, 50%, and 100%) of the total outer surface of concrete specimens (Figure 2), while in the case of vertically confined specimens, two ratios were adopted (25% and 50%). These configurations were applied to all concrete types. Finally, the percentages of CFRP wraps used were compared with the total cylinder’s surface, as presented in Table 5.

To easily distinguish between the various cylinders, the configurations were designated by the confinement rate, the orientation of the CFRP, and the concrete type. The first letter, T, H, or Q, indicates that the confinement was conducted totally (fully), half, or quarter. The second letter, C, indicates that the type of reinforcement is CFRP confinement. The third letter, L or T, means that the reinforcement orientation is longitudinal or transverse. Finally, the last letter, A, B, C, or D, signifies the ratio of water added to the concrete W/C being 0.33, 0.36, 0.401, or 0.522, respectively.

Prior to the compressive strength tests at 28 days under axial load, all concrete specimens were capped with sulfur mortar for accurate tests. After that, the specimens were tested under compressive and tensile strength (indirect tensile) tests to measure each type of reinforcement’s different strengths and behaviors. All tests were carried out using the MCC8 machine with 3000 kN capacity with an accuracy of ±1% (CONTROLS S.p.A, Liscate, Italy). The specimens were subjected to a stress rate of 0.5 MPa/sec until failure. Three Linear Variable Differential Transducers carried out strain acquisition (LVDT) placed at the mid-height of the specimens, as shown in Figure 3d.

## 4. Results and Discussion

### 4.1. Concrete Cylinders Stress Deformation

All the reinforced concrete cylinders (RCC) with CFRP had a typical behavior. Two phases were observed in Figure 4. This figure presents the trend curves of the tests carried out in triplicate. The first one presents an ascending phase with the rapid evolution of the stresses, while the second is defined by an ascending phase but with a slight slope due to the presence of a sufficient quantity of carbon fibers contributing to the compressive strength. In this case, it can be said that the volumetric ratio is sufficient for the reinforcement.

In the first phase, the load is essentially supported by the concrete. This is confirmed by the noise caused by the micro-cracks propagating towards the concrete core. Since the shrinking of the concrete during this phase is small, the lateral expansion of the cylinder must be too small, which can be explained by the fact that the fibers do not have a significant contribution in this stage.

In the second phase, the curves tend to flatten. Consequently, most of the deformations were supported by the fibers. As a result, the stresses slowly evolved until the configuration failed; the results obtained are depicted in Table 6. The reinforcement of cylinders by CFRP, whatever the fiber’s orientation and the volumetric ratio, improved the ultimate compressive stress and strain following:(1)$$\rho_{f}=\frac{A_{f}\pi D}{\left(\pi D^{2}\right)/4}$$
where
-*ρ_f_* is CFRP reinforcement ratio;-*A_f_* is the CFRP area;-*D* is the cylinder diameter.

### 4.2. Effect of Fiber Orientation

The obtained results of the reinforced concrete cylinders by CFRP according to the orientation of the reinforcements are presented in Figure 5 and Table 6. The transversely reinforced concrete cylinders (HCT and QCT) show higher stiffness than longitudinally reinforced cylinders (TCL, HCL, and QCL). The lateral expansion of concrete carried out the transfer of load to the fibers for the transversely reinforced cylinders. In contrast, in the cylinders that were longitudinally reinforced, the increase in the load as compared to the control cylinder is due to the following two reasons:The first is the cohesion brought by the fiber when added to the surface of the cylinder through the resin, thus forming the composite material of concrete and fiber that adhere to resist the compressive force.The second reason is that during the lateral expansion of concrete, a mechanism of stress transfer to the lateral direction of the fiber is created, and consequently, part of the load is taken by the latter.

From Figure 5, the longitudinally reinforced concrete cylinders can easily be deformed because the confining pressure of transversally reinforced cylinders is greater than that of the longitudinally reinforced cylinders induced to prevent deformations by the fiber that wraps up parts of the cylinder. Hence, when the stress of the cylinders approaches 90% of the ultimate compressive strength, the transverse deformation becomes very high due to the low contribution of the fiber, longitudinally oriented in the compressive strength, which affects the stiffness of the composite material. In this case, internal cracking progresses and leads to a rapid deformation of the reinforced concrete.

Furthermore, longitudinally reinforced concrete cylinders have a high ductility compared to transversely reinforced cylinders. This behavior can be useful during seismic loading because the plastic bearing can dissipate energy. The half and quarter transversely (HCT and QCT) confined specimens underwent an increase in the ultimate compressive stress from 3% to 15% and from 9% to 16%, respectively, compared to the half and quarter longitudinally confined cylinders (HCL and QCL). Nevertheless, a remarkable decrease in the ultimate compressive strain was recorded.

### 4.3. Concrete Strength Effect

Figure 6 investigates the effect of the compressive strength on the behavior of the confined concrete cylinders. Four concrete strengths were obtained (25.66, 30.27, 40.75, and 46.91 MPa).

The confinement efficiency for the configurations made with W/C ratios 0.33 and 0.36 was lower than that of ratios 0.401 and 0.522. When the stress reaches the ultimate elasticity, the cracks start propagating rapidly in the cylinders’ core, and the specimen’s failure occurs suddenly. The fully confined concrete specimens showed an increase in the ultimate compressive stress compared to the control specimens (31%, 32%, 55%, and 66% for W/C ratios of 0.33, 0.36, 0.401, and 0.522, respectively). The ultimate compressive stress was observed in the case of the half-confined specimens (HCT), i.e., 19%, 11%, 38%, and 52% for W/C ratios of 0.33, 0.36, 0.401, and 0.522, respectively. Meanwhile, the half confined longitudinally (HCL) of concrete specimens, an improvement in ultimate compressive stress of 24%, 23%, 58%, and 57% for W/C ratios of 0.33, 0.36, 0.401, and 0.522, respectively, were noted. The other two types of configurations (QCL and QCT) also showed an increase in the ultimate compressive stress (for QCL: 9%, 4%, 25%, and 28% and for QCT: 20%, 11%, 49%, and 48% for W/C ratios of 0.33, 0.36, 0.401, and 0.522, respectively).

### 4.4. Effect of Changing CFRP Volumetric Ratio

Figure 7 shows the change impact of the volumetric ratio and indicates that the increase in this ratio induces an increase in the ultimate compressive stress since the confining pressure depends essentially on the quantity of adopted CFRP.

An increase in ultimate compressive stress compared to control specimens from 3% to 66% was induced for W/C ratios ranging from 0.33 to 0.522 with a change in a volumetric ratio of 0.025, 0.01, and 0.0046. It is noticed that the strength gain of the confined concrete specimens improved inversely with the concrete compressive strengths. Gains in ultimate compressive stresses ranging from 11% to 57% were recorded for the half confined and quarter confined specimens for W/C ratios ranging from 0.33 to 0.522, through 0.36 and 0.401.

### 4.5. The Influence of Different Factors on the Tensile Strength

Table 7 summarizes the obtained tensile strengths by the indirect tensile strength test (Brazilian test) on cylindrical specimens (16 × 32 cm^2^) according to the standard NF P18-408 [37] for the various configurations.

A clear improvement in the tensile strength of the partially transversely confined (HCT and QCT) specimens was observed compared to the tensile strength of the control concrete specimen and the partially longitudinally confined specimens. However, this improvement becomes insignificant when comparing the tensile strength of specimens with different CFRP volumetric ratios. As it can be seen, there is an improvement of the tensile strength following the increase of the compressive strength, but it is not with the same magnitude.

### 4.6. Failure Mode

All longitudinally reinforced specimens failed by splitting (Figure 8a). Longitudinal cracks at the edges appeared along the length of the specimens, located mainly in the vicinity of the CFRP casting zone. The coalescence of the micro-cracks will form vertical macro-cracks splitting the specimen from top to bottom. This configuration causes tangential stresses around the reinforced area, and since the specimens have not been confined radially, the tangential stress is greater than the confining pressure, which induces this kind of failure. Kotsovos et al. [39] have defined splitting failure as the limiting case of a shear band failure with an inclination of 0° to the load axis.

Transversely reinforced specimens made from concrete with different W/C ratios of 0.36, 0.401, and 0.522 underwent a biconical fracture plane (Figure 8b), and micro-cracks due to micro-sliding at the paste/aggregate interface appeared. These propagate into the matrix following a cone formed above and below the aggregate, while the CFRP in these specimens has not been damaged. The failure of the specimens made with a 0.33 ratio of concrete and transversely reinforced was produced abruptly (Figure 8c) or by noticing the damage to the concrete and the CFRP with an explosive noise.

### 4.7. Statistical Modelling

#### 4.7.1. The Factorial Experimental Results

The summary of the measured compressive strength for both confinement modes, transversal ‘T’ or longitudinal ‘L’, at each experimental point is presented in Table 8.

The correlations between the predicted and observed values for both longitudinal confinement (LC) and transversal confinement (TC) are shown in Figure 9a,b. Again, it is seen that the points follow the fitting line. In addition, very high correlation coefficient and adjusted coefficient values were observed (Table 9), indicating a good correlation between the predicted and observed values.

#### 4.7.2. Variance Analysis

Table 10 presents the degrees of freedom, the sum of squares, the mean square, the F-ratio, and the probability. The latter confirmed that there is at least one significant effect in the model (Prob. > F) lower than 5%, regardless of the type of confinement.

#### 4.7.3. Longitudinal Confinement

The iso-response curves and the response surfaces (Figure 10) were used to determine the best W/C ratio and longitudinal confinement (LC) percentage to obtain the optimum compressive strength. From the response surfaces and the main effect plots (Figure 11a), it can be noted that an increase in the longitudinal confinement percentage LC (%) led to an increase in the compressive strength, while the increase in the W/C has a negative influence on the compressive strength.

According to Table 11, in which we present the results of the effect test, Prob. > |t| are lower than 0.05. This indicates that the W/C ratio and the longitudinal confinement percentage are considered the statistically insignificant interaction contrariwise. These results are confirmed by the interaction plots (Figure 11a), in which the intersection of the two lines indicates that the interaction of factors has no considerable effect on this response.

The mathematical relationship of the compressive strength for specimens with longitudinal confinement is given by:(2)$$\mathrm{CS}{\left(\mathrm{MPa}\right)}^{\prime }\;L^{\prime }=42.41-8.17\frac{\frac{W}{C}-0.426}{0.096}+7.68\frac{\mathrm{LC}\left(\%\right)-50}{50}+0.62\frac{\frac{W}{C}-0.426}{0.096}.\;\frac{\mathrm{LC}\left(\%\right)-50}{50}$$

#### 4.7.4. Transversal Confinement

Figure 12 shows the iso-response and surface response for compressive strength of transversal confinement (TC) specimens. The Iso-response plot shows that the compressive strength increases from nearly 39 MPa to 55 MPa with the transversal confinement percentage.

It is worth mentioning from the surface response that the increase in the transversal confinement percentage has increased the compressive strength, while the increase in the W/C ratio has led to its decrease.

Figure 13b shows the interaction plots. It is observed that the two lines do not intersect, thus indicating that interaction effects have no appreciable effect. These results are dependent on the estimated coefficients (Table 9) and the following relationship:(3)$$\mathrm{CS}{\left(\mathrm{MPa}\right)}^{\prime }\;T^{\prime }=41.56-8.35\frac{\frac{W}{C}-0.426}{0.096}+6.87\frac{\mathrm{TC}\left(\%\right)-25}{25}+1.06\frac{\frac{W}{C}-0.426}{0.096}.\;\frac{\mathrm{TC}\left(\%\right)-25}{25}\;$$

### 4.8. Effect of the Added Sand to the Resin on Reinforced Concrete Cylinders

In the context of confirming or refuting the obtained results reported by Abdulla et al. [15], in which the authors proposed the feasibility of adding a quantity of sand to the sealing materials up to a ratio of S/R equal to 1 to improve the mechanical and thermal characteristics of the resin, six specimens were strengthened with the same volumetric ratio of CFRP equal to 0.046, and in which we altered the sand content in the resin. The S/R ratios used were 0.50, 0.65, and 0.85. The results obtained are summarized in Table 12 and shown in Figure 14a.

The three obtained curves reveal the same behavior. Adding sand to the resin has increased the stiffness, the ultimate compressive stress, and the ductility of the reinforced concrete (RC) cylinders. The combination of sand and resin has created silanol bonds which are very strong and require high energy to break. Furthermore, by increasing the sand content in the sealing material, Young’s modulus of the mixture (sand/resin) increases, and the resin becomes increasingly rough. This increases tangential forces between the two surfaces, resin–concrete and resin–CFRP, and the adhesion between the surfaces becomes greater.

The specimens, which were sealed using the resin having S/R ratios of 0.50, 0.65, and 0.85, exhibited an increase in their ultimate stresses from 1.19% to 54.62%, while an improvement from 2.70% to 15.19% was observed in the ultimate strains. These will increase the absorption energy, making the element more ductile and reducing the cost of the resin used for fixing CFRP materials. Our results are in accordance with those of Abdulla et al. [15].

## 5. Conclusions

In conclusion, an experimental investigation of the compressive behavior of concrete cylinders externally confined by CFRP wrapping has been presented in this paper. Reinforcements (confinement) were performed using longitudinal, lateral, and radially spaced sheets. The effects of the principal variables, such as the concrete strength, the angle orientation, the volumetric ratio of CFRP, and the addition of sand to resin, were investigated. We made the following conclusions:

The transversely confined concrete cylinders led the CFRP to develop their capacity (strength) better than the longitudinal ones. However, all confined concrete specimens with a longitudinal orientation showed a larger flat region as a plastic region prior to failure than those with a transversal orientation. Such behavior may be very helpful for cyclic loading.

The configuration cylinders made with a low concrete strength showed a higher effect of CFRP confinement than those made with high concrete strength. Hence, the effect of CFRP confinement decreases with increasing the concrete strength.

Increasing the volumetric ratio of CFRP produced an increase in the compressive strength of the confined column from 3% to 56% compared to the control cylinder. Nevertheless, a decrease in the deformation was noticed.

The improvement in tensile strength was only recorded in the transversely confined specimens.

The results of the factorial plan showed a good correlation between the experimental and expected results.

The addition of sand to the resin increased the stress and the ultimate compressive strain, which confirms the results of Abdulla et al. [15]. Hence, this will increase the absorption energy and reduce the cost of the resin used for fixing CFRP materials.

It would be desirable to complete this study through further investigations on the problem associated with: (1) the durability of structural elements reinforced by CFRP in aggressive environments, and (2) the CFRPs compatibility issues encountered in historic masonry structures. It would also be useful to complete a comparative study between the reinforcement by CFRP and the FRCMs in order to present the advantages and disadvantages of FRCMs. The future works are also to calibrate with numerical modelling the obtained experimental results by using the Ansys software package.

## Figures and Tables

**Figure 1 materials-15-08232-f001:**
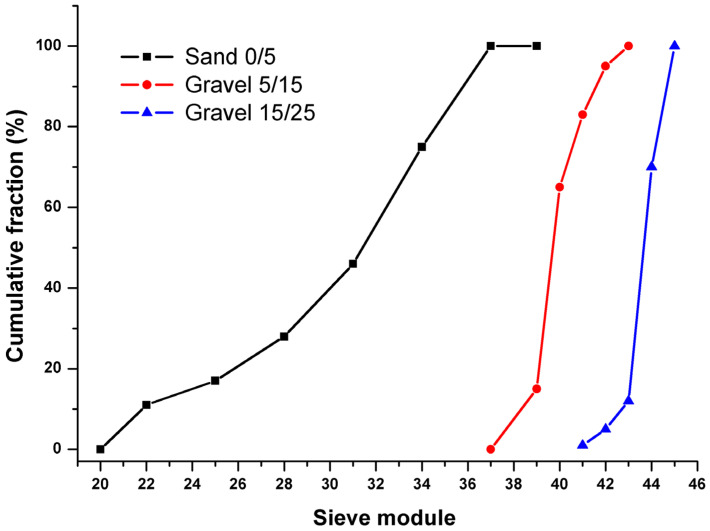
Grading curves of calcareous crushed aggregates according to NF P18-560.

**Figure 2 materials-15-08232-f002:**
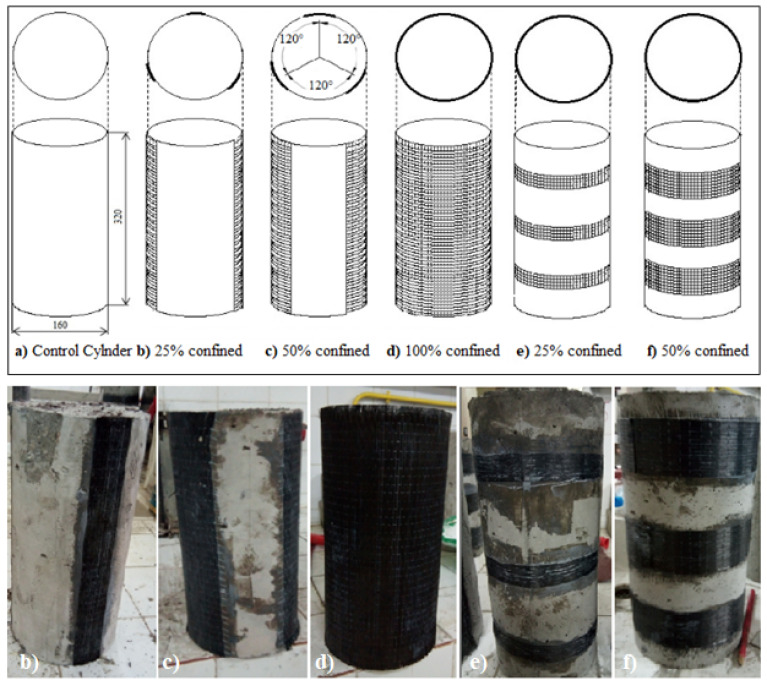
Various confinement (wrapping) configurations: (**a**) control cylinder CC; (**b**) quarter confined longitudinal QCL; (**c**) half confined longitudinal HCL; (**d**) totally confined longitudinal TCL; (**e**) quarter confined transversal QCT; and (**f**) half confined transversal HCT.

**Figure 3 materials-15-08232-f003:**
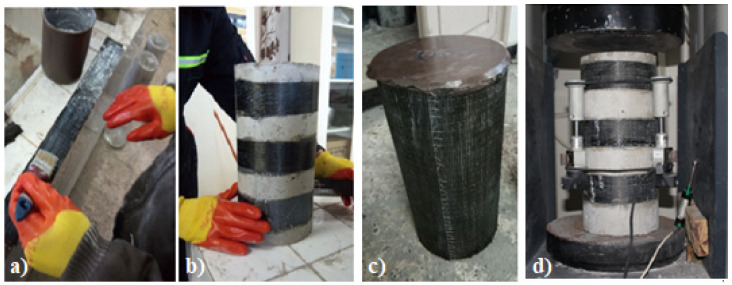
(**a**) Epoxy adhesive application; (**b**) specimens wrapping; (**c**) capping of concrete cylinders; and (**d**) test instrumentations.

**Figure 4 materials-15-08232-f004:**
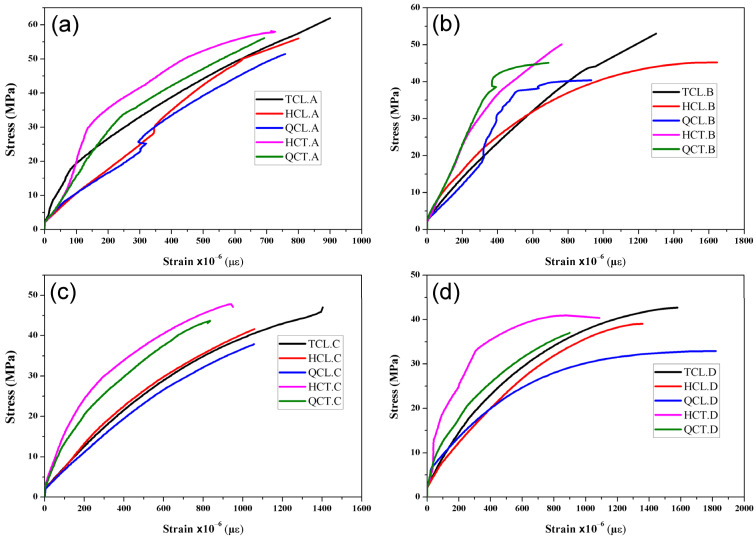
Stress–strain curves under monotonic load: (**a**) concrete made with 0.33 W/C; (**b**) concrete made with 0.36 W/C; (**c**) concrete made with 0.401 W/C; and (**d**) concrete made with 0.522 W/C.

**Figure 5 materials-15-08232-f005:**
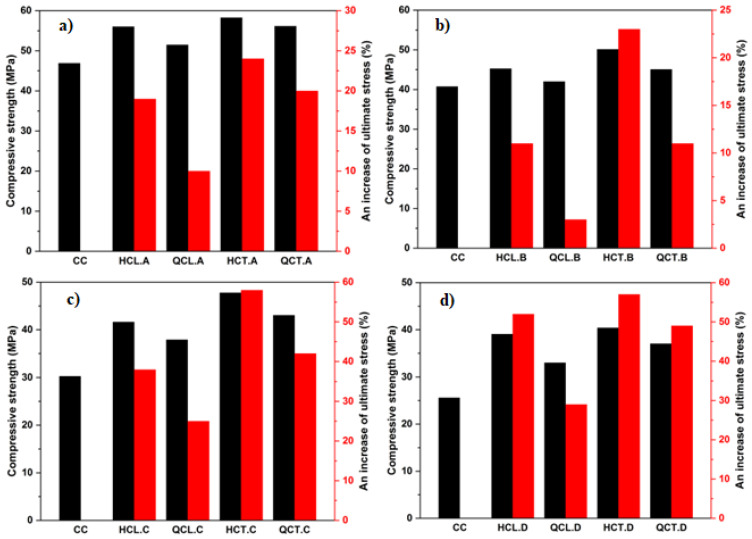
The effect of CFRP orientation: (**a**) concrete made with 0.33 W/C; (**b**) concrete made with 0.36 W/C; (**c**) concrete made with 0.401 W/C; and (**d**) concrete made with 0.522 W/C.

**Figure 6 materials-15-08232-f006:**
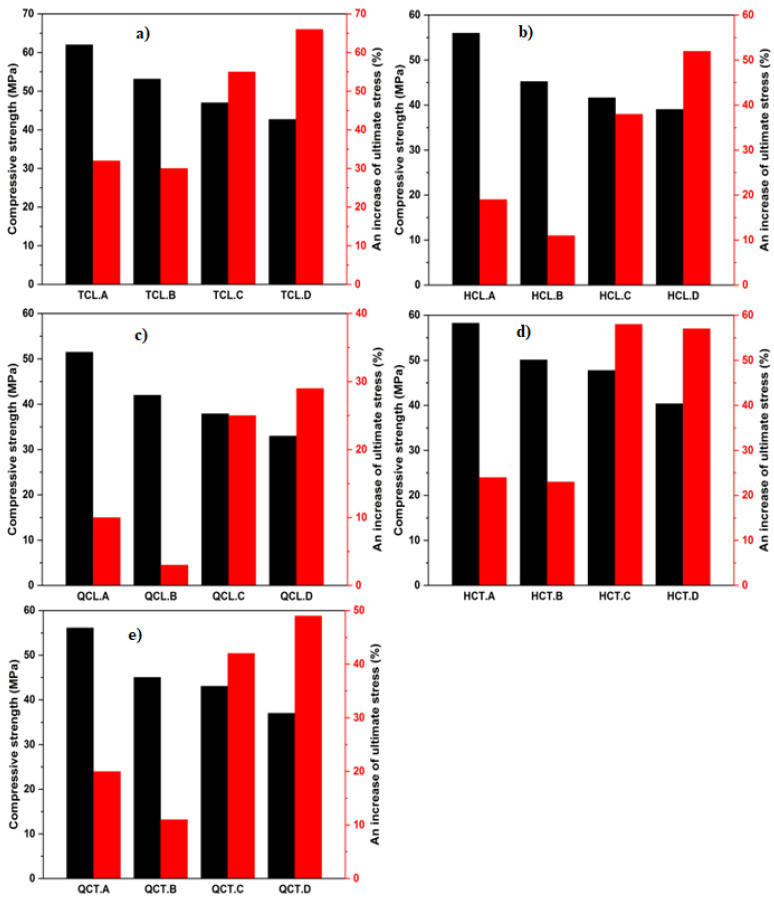
Effect of changing concrete strength: (**a**) TCL configuration; (**b**) HCL configuration; (**c**) QCL configuration; (**d**) HCT configuration; and (**e**) QCT configuration.

**Figure 7 materials-15-08232-f007:**
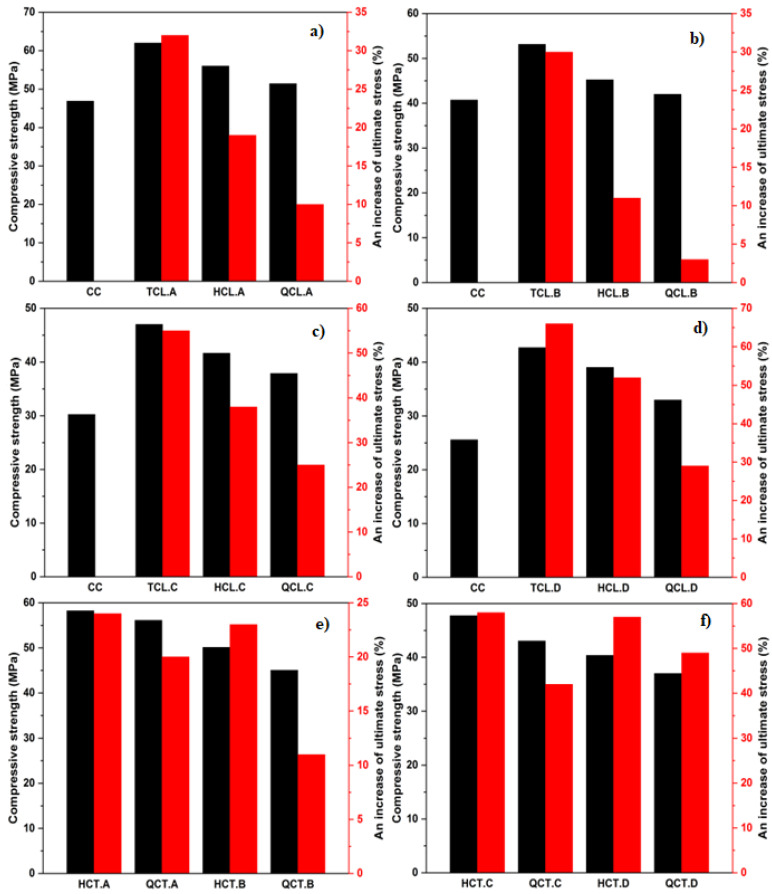
Volumetric ratio effect: (**a**) longitudinal confinement of 0.33 W/C ratio; (**b**) longitudinal confinement of 0.36 W/C ratio; (**c**) longitudinal confinement of 0.401 W/C ratio; (**d**) longitudinal confinement of 0.522 W/C ratio; (**e**) transversal confinement of 0.33 and 0.36 of W/C ratios; and (**f**) transversal confinement of 0.401 and 0.522 of W/C ratios.

**Figure 8 materials-15-08232-f008:**
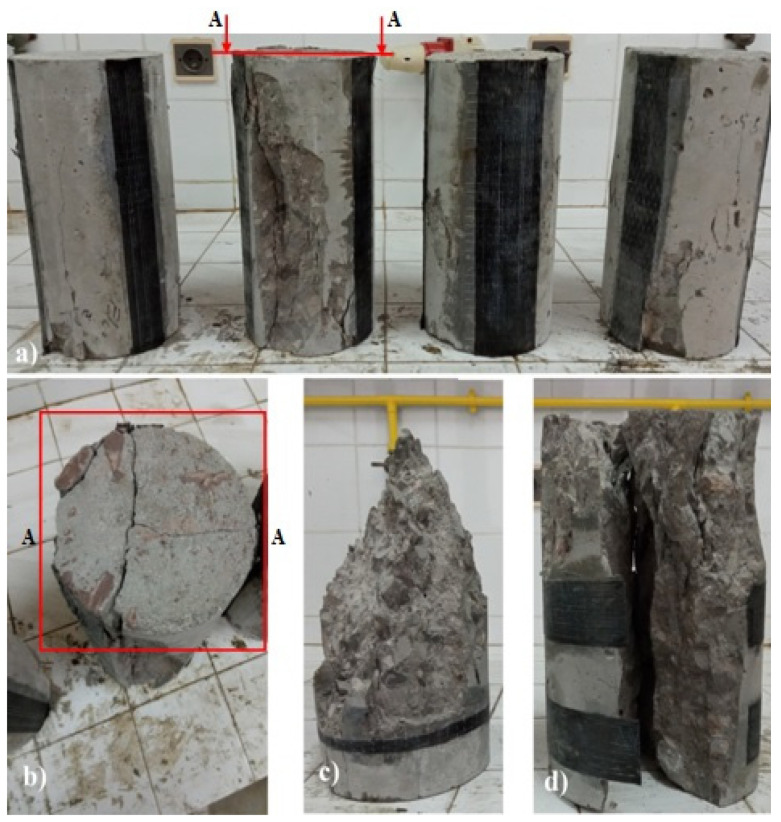
Failure modes of plain concrete cylinders after compression tests: (**a**) failure by splitting; (**b**) cross-sectional area of first failure mode; (**c**) shear (cone); and (**d**) brutal failure of concrete.

**Figure 9 materials-15-08232-f009:**
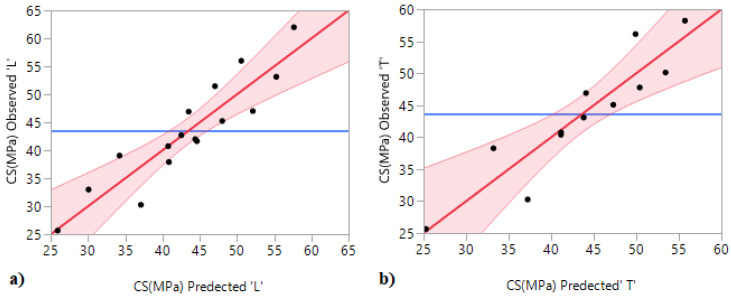
Correlation between experimental and predicted values: (**a**) longitudinal confinement (LC); and (**b**) transversal confinement (TC).

**Figure 10 materials-15-08232-f010:**
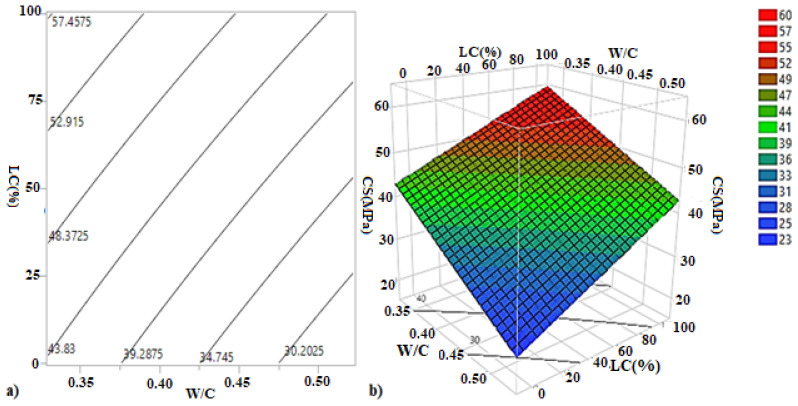
(**a**) Iso-response curves and (**b**) response surfaces of compressive strength of the longitudinal confinement.

**Figure 11 materials-15-08232-f011:**
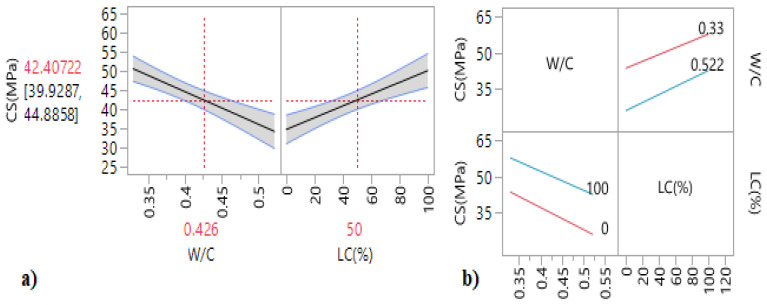
(**a**) Main effect plots and (**b**) interaction plots of the compressive strength of the longitudinal confinement.

**Figure 12 materials-15-08232-f012:**
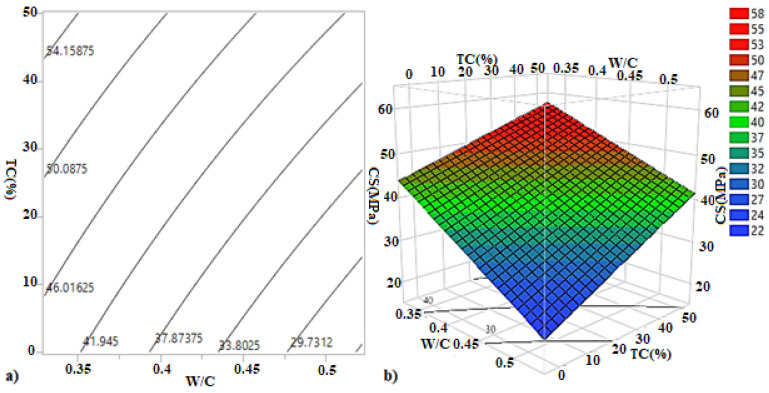
(**a**) Iso response curves and (**b**) response surfaces of the compressive strength of the transversal confinement.

**Figure 13 materials-15-08232-f013:**
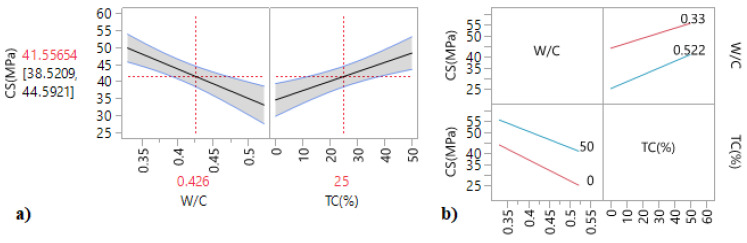
(**a**) Main effect plots and (**b**) interaction plots of the compressive strength of the transversal confinement.

**Figure 14 materials-15-08232-f014:**
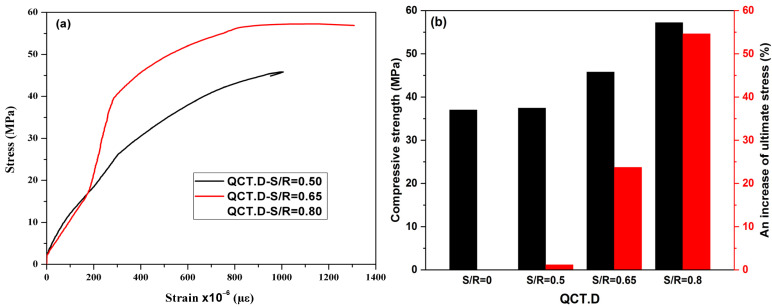
Effect of adding sand to resin: (**a**) stress–strain curve; and (**b**) compressive strength.

**Table 1 materials-15-08232-t001:** A review of previous studies on concrete reinforcement using CFRP composites.

Authors	Compressive Strength (MPa) and Fiber Orientation (°)	Studied Parameters	Selected Results
Liang et al. [16]	34.2/0°	Slenderness ratio, CFRP layer number. Width of FRP bands.	Increasing the number of CFRP layers for strengthening columns increases compressive strength and stiffness. The reverse was when the strip spacing and slenderness ratio were increased.
Liang et al. [17]	20/0°	Slenderness ratio. CFRP layer number.	The axial load capacity of the columns increases with the number of CFRP layers. However, the initial stiffness of the columns does not always increase with the number of CFRP layers.
Babba et al. [18]	20/0°	The CFRP distribution on the cylinders.	Using CFRP with small widths on the cylinder considerably improves the carrying capacity and ductility of the cylinder.
Ismail et al. [19]	47.19/0°, +20°	Spacing and location of CFRP (horizontal and helical).	The confined concrete cylinders horizontally present a higher ultimate compressive strength at about 18.2% compared to those helical reinforced.
Benzaid et al. [20]	25.93/0°	The reinforcement ratio.	Increasing the reinforcement ratio for columns has increased the resistance to failure with a decrease in deformation.
Yin et al. [21]	30.6/0°	The type thickness of CFRP and the spacing of transversal spiral bars.	The increase in the reinforcement ratio induces a decrease in the concrete expansion. This is expressed by the confinement pressure which was exercised on the CFRP.
Belouar et al. [22]; Chikh et al. [23]; Benzaid et al. [24]	25.93–61.83/0°	The confinement was carried out with a variation of the compressive strength.	The confinement efficiency decreases with the increase in concrete strength.
Antonio De Luca et al. [25]	34/0°	Cross section.	The cross-section geometrical configuration affects the confinement efficiency. For example, the effectiveness is better for a squared disposition than a rectangular one.
Sadeghian et al. [26]	30/0°, ±45°, and 90°	Orientation and width of CFRP bands.	The confined concrete cylinders with a CFRP transversal orientation have a bilinear behavior. The plastic zone is more important when the fiber’s angle points towards 45°.
Li et al. [27]	40/0°, 45°, and 90°	Orientation of GFRP.	The ultimate compressive stress, the ductility, and the failure mode of concrete cylinders fully confined with GFRP depend on the reinforcement thickness and the fiber’s orientation. The orientation of the fibers with an angle greater than 0° induces a decrease in the ultimate compressive stress.
Parvin & Jamwal [28]	20 to 40/0°, ±15°	CFRP thickness. Concrete strength.	The strength gain for FRP confined cylinders mainly depends on the initial compressive strength and the fiber orientation.
Rochette et al. [29]	40/0°, ±15°	Columns section. The reinforcement ratio.	All specimens were fully confined; the reinforced specimens with a 0° orientation angle have higher stiffness and ultimate compressive stress than those reinforced at an angle of 15°.
Mirmiran et al. [30]	31/±15°	The thickness of GFRP type.	The increase in GFRP thickness induces an increase in ductility, stiffness, compressive strength, and ultimate deformation.

**Table 2 materials-15-08232-t002:** Physico-chemical properties of the used cement.

Compound	SiO_2_	Al_2_O_3_	Fe_2_O_3_	CaO	MgO	SO_3_	K_2_O + Na_2_O
(%)	27.83 ± 0.11	6.21 ± 0.14	3.12 ± 0.03	57.22 ± 0.31	0.94 ± 0.06	2.02 ± 0.02	0.14 ± 0.03
Setting time		Initial		2 h and 50 min			
		Final		4 h and 06 min			
C3S = 56.60% C2S = 22.98% C3A = 11.18% C4AF = 9.48%		- Ignition loss L.O.I. = 2.41 according to NF EN 196-2 - Specific Surface Area S.S.A. = 3891 cm^2^/g - Density = 3150 kg/m^3^ - Expansion of cement = 1.25 mm					

**Table 3 materials-15-08232-t003:** Mixtures’ design and main properties of concrete types.

Components and Characteristics of Concrete Mixtures	Concrete Type 1	Concrete Type 2	Concrete Type 3	Concrete Type 4
Crushed Sand (0/5) (kg/m^3^)	540	540	540	540
Gravel 5/15 (kg/m^3^)	420	420	420	420
Gravel 15/25 (kg/m^3^)	850	850	850	850
Cement CEM II42.5 (kg/m^3^)	400	400	400	400
Water (kg/m^3^)	132	144	160.4	208.8
Water/Cement ratio	0.33	0.36	0.401	0.522
Superplasticizer (%)	1.8	1.8	1.6	0
Slump value (cm)	7	8	8	12
Workability	S2	S2	S2	S3
Compressive strength (MPa)	46.91	40.75	30.27	25.66
Tensile strength (MPa)	3.15	3	2.52	2.17
S2: Plastic concrete S3: Highly plastic concrete				

**Table 4 materials-15-08232-t004:** Properties of unidirectional carbon fiber wraps and the epoxy adhesive.

Type of CFRP/Adhesive	Tensile Strength (MPa)	Elastic Modulus (MPa)	Elongation at Break (%)	Thickness (mm)	Density
CarboDur^®^ SikaWrap^®^	4000	23,000	1.7	0.129	1.82
Sikadur 330	30	4500	0.9	/	1.3

**Table 5 materials-15-08232-t005:** Percentages of CFRP wraps used as compared to the cylinder’s surface total.

Concrete Type	Horizontal Orientation of CFRP				Vertical Orientation of CFRP	
	0%	25%	50%	100%	25%	50%
	Control Concrete Cylinders	of the total surface is confined by three equidistant widths (120°)		of the total surface is confined	of the total surface is confined by three equidistant widths	
Type 1	5 samples	5 samples	5 samples	5 samples	5 samples	5 samples
Type 2	5 samples	5 samples	5 samples	5 samples	5 samples	5 samples
Type 3	5 samples	5 samples	5 samples	5 samples	5 samples	5 samples
Type 4	5 samples	5 samples	5 samples	5 samples	5 samples	5 samples

**Table 6 materials-15-08232-t006:** Specimens’ experimental results of the compressive strength.

Series	Concrete Mixture	Ultimate Compressive Stress (MPa)	Increase (%)	Volumetric Ratio *ρ_f_*
CC	Type 1	46.91		0
TCL.A		62.00	32	0.025
HCL.A		56.00	19	0.01
QCL.A		51.46	10	0.0046
HCT.A		58.23	24	0.01
QCT.A		56.13	20	0.0046
CC	Type 2	40.75		0
TCL.B		53.15	30	0.025
HCL.B		45.25	11	0.01
QCL.B		42.00	03	0.0046
HCT.B		50.12	23	0.01
QCT.B		45.08	11	0.0046
CC	Type 3	30.27		0
TCL.C		47.02	55	0.025
HCL.C		41.64	38	0.01
QCL.C		37.91	25	0.0046
HCT.C		47.78	58	0.01
QCT.C		43.08	42	0.0046
CC	Type 4	25.60		0
TCL.D		42.71	66	0.025
HCL.D		39.06	52	0.01
QCL.D		33.00	29	0.0046
HCT.D		40.39	57	0.01
QCT.D		37.02	49	0.0046

**Table 7 materials-15-08232-t007:** Specimens’ experimental results of tensile strength.

Water/Cement Ratios	Indirect Tensile Strength (MPa)					
	CC	TCL	HCL	QCL	HCT	QCT
0.33	3.15	3.33	3.12	3.10	4.06	3.95
0.36	3.00	3.07	2.88	2.97	3.11	3.01
0.401	2.52	2.49	2.50	2.48	2.75	2.68
0.522	2.17	2.22	2.04	2.15	2.62	2.64

**Table 8 materials-15-08232-t008:** Factorial experimental results.

Water/Cement Ratios	Confinement (%)	Compressive Strength (MPa) Longitudinal Confinement	Compressive Strength (MPa) Transversal Confinement
0.33	100	62.00	/
0.36	100	53.15	/
0.401	100	47.02	/
0.522	100	42.71	/
0.33	50	56.00	58.23
0.36	50	45.25	50.12
0.401	50	41.64	47.78
0.522	50	39.06	40.39
0.33	25	51.46	56.13
0.36	25	42.00	45.08
0.401	25	37.91	43.08
0.522	25	33.00	38.27
0.33	0	46.915	46.915
0.36	0	40.75	40.75
0.401	0	30.27	30.27
0.522	0	25.66	25.66

**Table 9 materials-15-08232-t009:** Fitting results summary.

	Longitudinal Compressive Strength (MPa)	Transversal Compressive Strength (MPa)
Coefficient of Determination R^2^	0.837256	0.847243
Adjusted R^2^	0.79657	0.789959
Root Mean Square Error (RMSE)	4.279522	4.34963
Mean of Response	43.42469	43.55625

**Table 10 materials-15-08232-t010:** Variance analysis (ANOVA) of derived models.

	Source	Freedom Degree	Sum of Squares	Mean Square	F-Ratio
Longitudinal Compressive Strength (MPa)	Model	3	1130.6432	376.881	20.5785
	Error	12	219.7717	18.314	Prob. > F
	Total	15	1350.4149		<0.0001 *
Transversal Compressive Strength (MPa)	Model	3	839.46144	279.820	14.7902
	Error	8	151.35422	18.919	Prob. > F
	Total	11	990.81566		0.0013 *

*: (Prob. > F) lower than 5%.

**Table 11 materials-15-08232-t011:** Effect test.

	Model Term	Estimation	Standard Error	T Ratio	Prob. > |t|
Longitudinal Compressive Strength (MPa)	Constant	42.407223	1.137562	37.28	<0.0001 *
	W/C	−8.179192	1.425854	−5.74	<0.0001 *
	LC (%)	7.6852841	1.516749	5.07	0.0003 *
	W/C × LC (%)	0.6186268	1.901139	0.33	0.7505
Transversal Compressive Strength (MPa)	Constant	41.55654	1.316388	31.57	<0.0001 *
	W/C	−8.346614	1.650001	−5.06	0.0010 *
	TC (%)	6.8695833	1.61224	4.26	0.0028 *
	W/C × TC (%)	1.06	2.020831	0.52	0.6141

*: (Prob. > |t|) lower than 5%.

**Table 12 materials-15-08232-t012:** Experimental results of compressive strength.

Series	Sand/Resin Ratio	$\sigma_{u}(\mathrm{MPa})$	Increase (%)	$\epsilon_{u}(µɛ)$	Increase (%)
QCT.D	0	37.02		898.92	
QCT.D	0.5	37.46	1.19	923.2	2.70
QCT.D	0.65	45.81	23.74	991.8	10.33
QCT.D	0.80	57.24	54.62	1035.5	15.19

## Data Availability

Data sharing not applicable.

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
