# Peer review of "External Confined Concrete Cylinders Behavior under Axial Compression Using CFRP Wrapping"

_materials, 2022, doi:10.3390/ma15228232_

Round 1

Reviewer 1 Report

In the present paper, effects of carbon fiber polymer sheet on compressive strength of small concrete cylinders. The sheets were adhesive to the surface of the cylinder in horizontal and vertical direction. The ratio in area of covered by CFRP sheet to total side area of the cylinder is 0, 25%, 50% and 100%. Four concretes with water to binder ratio of 0.33, 0.36, 0.40 and 0.52 were used as strengthen samples. Some specific comments on the paper are listed below.        

1) As the CFRP sheet was partially use on the side surface of the cylinder, in the view of compressive strength enhancement, how can prevent of failure just at the place where there are no CFRP sheet? How can you confirm the reliability of the strengthening? This issue needs to be described in the test program.

2) Test result does indicates the increase on compressive load carrying capacity of the cylinder after used of CFRP sheet, please showing and explain the differences before and after use of CFRP sheet.

3) Figure 4, more than one cylinder on each strengthen manner are required to check the reliability of the method.

Reviewer 2 Report

In this manuscript, the authors investigate the behavior of confined concrete cylinders under uniaxial compression and Brazilian testing by using the Carbon Fiber Reinforced Polymer Composite and varying parameters such as concrete strength, angle orientation, and the volumetric ratio. This is an interesting manuscript with useful results. However, there are number of problems that have to be addressed before this manuscript can be published. List of my comments/suggestions is below:

1.      Line 29: Space before 31% is missing.

2.      Line 160: It is more correct to say “stress rate” rather than “load rate” if units are MPa/sec (this is only if the authors really used stress rate and not the load rate).

3.      Figure 4 provides stress-strain curves; however, the y-axis indicates that this is compressive strength. Strength is a single point at the top, so authors have to change axis name. In addition, what exactly the authors mean my micro strain? What are the units on the x-axis? Is it strain * 10^-6?

4.      Line 179: Perhaps “small” is better to use than “slight” in this sentence? Or, perhaps, a different word?

5.      Figure 6: I suggest that the authors improve the quality of this figure. It is hard to read y-axis labels as it gets blurry.

6.      Figure 6: Another comment is regarding using the word “increasement”. This is a very old and rare word that is not used nowadays. I suggest that the authors replace it with “an increase”.

7.      Figure 6: What do authors mean by ultimate stress? Perhaps ultimate strength (more common)? Please, also specify whether this is compressive or tensile ultimate stress. It is ok to leave the word stress if the authors prefer this way but use it as ultimate tensile/compressive stress.

8.      Line 159: Since the authors indicated max capacity I suggest that they also include the accuracy of the load cell.

9.      I suggest that the authors mention the dimensions of their specimens somewhere in the method/experimentation/materials section: length and diameter.

10.  No information is provided on boundary conditions for uniaxial compression and the Brazilian test. I suggest that the authors indicate what they used as a boundary condition in their experiments in the Methods section and briefly explain why. In addition, in my opinion, the authors also have to briefly mention the effect of boundary conditions as this is essential for understanding. I will explain below why I think boundary conditions are important.

The reason why this is important is because if we use a rough surface at the loading platen as a boundary, for example, fine grained sandpaper, this increases friction and resists the lateral expansion of the sample, resulting in a confining stress near the loading platen. In addition, placing a rough surface as a boundary condition, can change the mode of failure. For example, if the mode was axial splitting under uniaxial compression, a rough boundary will suppress this mode of failure. On the other hand, in order to minimize lateral confinement near the loading platens, a friction-free boundary should be placed between the sample and the loading platens, for example, polyethylene sheets. Generally, in order to obtain uniaxial stress state during loading, friction-free boundary is used. On contrary, when boundary with friction is used, we get a triaxial state of stress near the loading platens and that is why it is required to have a sufficiently long sample to ensure uniaxial state of stress in its middle part. It seems that the authors did have long enough samples, which is good.

The following references have to be added to support this argument (in these papers the authors used both sand paper as rough boundary condition, first paper, and thin polyethylene sheets, second paper, in order to observe the difference in the behavior; the authors also provided analysis on the observed results):

Renshaw, C. E., Schulson, E. M., Iliescu, D., & Murdza, A. (2020). Increased Fractured Rock Permeability After Percolation Despite Limited Crack Growth. Journal of Geophysical Research: Solid Earth, 125(8), 1–10. https://doi.org/10.1029/2019JB019240

Renshaw, C. E., Murdza, A., & Schulson, E. M. (2021). Experimental Verification of the Isotropic Onset of Percolation in 3D Crack Networks in Polycrystalline Materials With Implications for the Applicability of Percolation Theory to Crustal Rocks. Journal of Geophysical Research: Solid Earth, 126(12), 1–9. https://doi.org/10.1029/2021JB023092

11.  In the introduction, perhaps at the end, I suggest that the authors add a sentence or two and formulate their goals a bit more specifically, i.e. no just saying that the goal is to investigate the behavior of such samples with such things, but also adding why this is important, what practical problems this will potentially solve/improve understanding.

Reviewer 3 Report

This research topic is at least 25 years old and has already been extensively studied in all its aspects. The authors cite an absolutely insufficient number of articles, which does not do justice to the intense research activity carried out on the subject.

Experts in the field are very familiar with the behavior of cylindrical concrete specimens wrapped in CFRP and this article adds little new to the scientific literature. The only real new data concerns the addition of sand to the resin in the monoaxial test on CFRP-wrapped cylinders (while its effect is already known in the beams). This is too little to justify publication in an international journal, especially in light of the considerations made below.

In fact, it is worth noting that the initial euphoria with regard to fiber-reinforced materials, in particular with carbon fibers, has gradually faded in the face of the environmental sustainability problems that these materials pose. Furthermore, also from the structural point of view the CFRPs pose numerous critical issues. In fact, we must always keep in mind that reinforcing a single structural element significantly modifies the interaction that this structural element has with the rest of the structure. Therefore, an experimentation on the behavior of the single structural element is not conclusive. In the specific case of structural elements wrapped in CFRP, it is well known that the strong increase in stiffness of the wrapped element triggers criticalities on the neighboring structural elements. In essence, the structural vulnerability moves from the reinforced element to adjacent elements and is not resolved as a whole. More than 25 years since the beginning of the study of these materials, this aspect is widely known in the sector, while the authors seem to ignore it. Finally, the CFRPs pose compatibility problems with historical masonry structures.

The unresolved problems on the two fronts of structural safety (first) and environmental sustainability (later) have led researchers to gradually shift their attention to FRCMs. What the reviewer feels like advising the authors is to devote their energies to experimenting with FRCMs and to use the results presented in this article as a comparison, starting from which to highlight the advantages and disadvantages of FRCMs.

Round 2

Reviewer 1 Report

The revised version can be accepted for publication in the special issue now.

Reviewer 2 Report

The authors provided clear answers/responses to all the reviewer's comments.

Reviewer 3 Report

Since version 2 of the article is essentially unchanged, this reviewer’s final decision also does not change.

Authors should understand that a reviewer does not enjoy rejecting an article and should trust more suggestions made to them. If the reviewer argues that this is an old topic, already widely explored, he/she evidently has valid reasons to say so. If the Authors had used the time between the two revisions to do a thorough literature search on the subject, they could have easily found similar experimental works, conducted over 25 years ago.